# BAP31 Knockout in Macrophages Affects CD4^+^T Cell Activation through Upregulation of MHC Class II Molecule

**DOI:** 10.3390/ijms241713476

**Published:** 2023-08-30

**Authors:** Bo Zhao, Lijun Sun, Qing Yuan, Zhenzhen Hao, Fei An, Wanting Zhang, Xiaoshuang Zhu, Bing Wang

**Affiliations:** Institute of Biochemistry and Molecular Biology, College of Life and Health Sciences, Northeastern University, Shenyang 110819, China; 1810068@stu.neu.edu.cn (B.Z.); 1801357@stu.neu.edu.cn (L.S.); 1610069@stu.neu.edu.cn (Q.Y.); 1710065@stu.neu.edu.cn (Z.H.); 2201435@stu.neu.edu.cn (F.A.); 2110482@stu.neu.edu.cn (W.Z.); 2010494@stu.neu.edu.cn (X.Z.)

**Keywords:** BAP31, CD4^+^T cell, macrophages, MHC-II, activation, differentiation

## Abstract

The differentiation of CD4^+^T cells is a crucial component of the immune response. The spleen and thymus, as immune organs, are closely associated with the differentiation and development of T cells. Previous studies have suggested that BAP31 may play a role in modulating T cell activation, but the specific impact of BAP31 on T cells through macrophages remains uncertain. In this study, we present evidence that BAP31 macrophage conditional knockout (BAP31-MCKO) mice display an enlarged spleen and thymus, accompanied by activated clustering and disrupted differentiation of CD4^+^T cells. In vitro co-culture studies were conducted to investigate the impact of BAP31-MCKO on the activation and differentiation of CD4^+^T cells. The examination of costimulatory molecule expression in BMDMs and RAW 264.7 cells, based on the endoplasmic reticulum function of BAP31, revealed an increase in the expression of antigen presenting molecules, particularly MHC-II molecule, in the absence of BAP31 in BMDMs or RAW264.7 cells. These findings suggest that BAP31 plays a role in the activation and differentiation of CD4^+^T cells by regulating the MHC class II molecule on macrophages. These results provide further support for the importance of BAP31 in developing interaction between macrophages and CD4^+^T cells.

## 1. Introduction

Macrophages, being important antigen-presenting cells, engulf foreign antigens through phagocytosis or infection, and their activation is facilitated by the interaction with PRRs and TLRs, leading to the upregulation of costimulatory molecules and the presentation of antigenic peptides via MHC class I or II molecules to naïve T cells [1,2]. Macrophages possess various functions, including phagocytosis, antigen presentation, and the secretion of molecules such as growth factors, cytokines, complement components, and prostaglandins [3]. As important components of both the innate and adaptive immune systems, macrophages play a crucial role in the activation of T cells [4]. Specifically, the MHC class II molecule on macrophages serve as “signal 1” for the direct activation of T helper (Th) cells, as they are recognized by the TCR and bound with the assistance of the CD4 co-receptor [5]. The activation of T cells requires further activation signals, such as costimulatory molecules present on macrophages. Costimulatory molecules are commonly referred to as the second signal of T cell activation. The absence of stimulation from costimulatory molecules can result in T cell apoptosis or cell cycle arrest [6]. Additionally, the activation of macrophage-derived cells necessitates the presence of cytokines, which are known as the third signal [4]. Under the joint action of various signal molecules, naïve T cells proliferate and differentiate into different T cell subsets, each fulfilling their respective functions. Activated macrophages exhibit elevated levels of costimulatory and antigen-presenting molecules, such as CD80, CD86 and MHC class I and II molecules [7]. Macrophages serve as both antigen-presenting cells (APCs) and producers of cytokines that can influence the development of T cell responses. Additionally, macrophages are significant contributors of costimulatory molecules, which are essential for the effective activation of T cells. The expression of these co-stimulators is largely regulated by cytokines [8].

T lymphocytes undergo maturation in the thymus and subsequently migrate to peripheral lymphoid organs, such as the spleen, for further differentiation. CD4^+^T cells are particularly crucial in adaptive immunity against diverse pathogens [9]. The main sub-sets of CD4^+^T cells, namely Th1, Th2, and Th17 cells, undergo differentiation based on the cytokines they receive upon interaction between their T cell receptors and antigens [10,11]. Th1 cells, induced by IL-12 and IFN-γ, are responsible for host defense against intracellular pathogens through the specific expression of IFN-γ. Th2 cells, induced by IL-4, secrete IL-4, IL-5, and IL-13 to protect the host from worm invasion [12]. Additionally, the combination of IL-6 and TGF-β induces the expression of IL-17 in Th17 cells, a subset of helper T cells that have been subsequently identified and are known to have a significant role in the elimination of extracellular pathogens [13].

BAP31, a 28 kDa integral membrane protein located in the endoplasmic reticulum [14,15,16], forms a heterooligomer complex with a closely related protein called bap29 [14,17]. BAP31 is involved in endoplasmic reticulum retention and/or withdrawal of membrane-bound IgD [18], cystic fibrosis transmembrane conductance regulator [19], tetrapeptide [20], major histocompatibility complex class I protein [21] and cell recombinant protein within the endoplasmic reticulum [16]. There is evidence suggesting that BAP31 contributes to the structural integrity of the endoplasmic reticulum membrane. The role of BAP31 in endoplasmic reticulum membrane structure has been suggested due to its association with actin and myosin, which are cytoskeletal components [22]. Recent studies have revealed that BAP31 is a multifunctional protein involved in various immune cellular functions. Our previous research has demonstrated the involvement of BAP31 in T cell activation through the T cell antigen receptor (TCR) signaling pathway [23]. Furthermore, BAP31 has been shown to impact macrophage polarization by regulating the activation of helper T cells [24]. Additionally, BAP31 plays a role in regulating IRAK1-dependent neuroinflammation in microglia [25]. However, there is currently limited understanding regarding the BAP31 knockout in macrophages on the activation of CD4^+^T cells. Therefore, the objective of this study was to investigate the role of BAP31 knockout in macrophages on the activation of CD4^+^T cells by employing a knockdown approach in bone marrow-derived macrophages (BMDMs) and RAW264.7 cells.

## 2. Results

### 2.1. BAP31 Is Involved in the Development of Spleen and Thymus in BAP31-MCKO Mice

To investigate the impact of BAP31 on immune function in mice, we procured BAP31^flox/flox^Lyz2-cre (BAP31^−/−^) mice through conditional knockout techniques. To generate bone marrow-derived macrophages (BMDMs) derived from BAP31 conditional knockout mice, we utilized the supernatant of L929 cell-cultivated mouse bone marrow cells for a duration of 7 days to induce the formation of BMDMs. The purity of BMDMs (CD11b^+^F4/80^+^) from BAP31^flox/flox^ (BAP31^+/+^) and BAP31^−/−^ mice examined using flow cytometry was over 90% (Figure 1A). Additionally, we evaluated the efficiency of BAP31 protein knockout in BMDMs obtained from BAP31^+/+^ and BAP31^−/−^ mice through Western blot analysis (Figure 1B). The expression levels of BAP31 protein in the BMDMs of BAP31^−/−^ mice exhibited a reduction exceeding 90% compared to BAP31^+/+^ mice (Figure 1B). Simultaneously, the efficacy of BAP31 knockdown was assessed through quantitative RT-PCR analysis of RNA levels; the expression levels of BAP31 RNA in the BMDMs of BAP31^−/−^ mice exhibited a reduction of over 70% compared to BAP31^+/+^ mice (Figure 1C). The spleen and thymus of BAP31^−/−^ mice displayed significantly larger sizes and weights compared to BAP31^+/+^ mice (Figure 1D). Thymuses and spleens were isolated, and the main immune cell subsets were determined by flow cytometry. In comparison to BAP31^+/+^ mice, BAP31^−/−^ mice displayed normal proportions and absolute numbers of CD8 thymocytes, as well as normal proportions and absolute numbers of splenic B cells (B220^+^) and dendritic cells (CD11c+ DCs) (Appendix A). Overall, BAP31-MCKO had no discernible impact on the development of CD8 T cells, B cells, and dendritic cells, but did affect the frequency of CD4 T cells. Based on the above observations, we evaluated the expression of molecules related to CD4^+^T cell activation and differentiation in the spleen and thymus under physiological conditions. Total RNA was isolated from the spleen and thymus of both BAP31^+/+^ and BAP31^−/−^ mice and further subjected to quantitative RT–PCR. BAP31 knockout was found to induce the activation of T cells in the spleen and thymus (Figure 1E). The conditional knockout of BAP31 resulted in a significant decrease in the Th1 and Th2 populations, while exhibiting an increase in the Th17 subset (Figure 1F–H). 

### 2.2. BAP31-MCKO Facilitates CD4^+^T Cell Activation

Bone marrow-derived macrophages (BMDMs) are primary macrophage cells that are derived from bone marrow cells with the aid of growth factor in vitro [26]. Naïve CD4^+^T cells, co-cultured with macrophages obtained from wild-type mouse spleen cells through magnetic bead sorting, exhibited a high purity of sorted cells (CD3^+^CD4^+^) of up to 97.8%, as detected by flow cytometry. These cells could be utilized in subsequent experiments (Appendix A). Bone marrow-derived macrophages were divided into control, LPS, and IL-4 groups, and were separately co-cultured with naïve CD4^+^T cells at a ratio of 1:10 for an additional 72 h in direct-contact co-culture conditions. Naïve CD4^+^T cells co-cultured with BMDMs without any stimulation were used as control. Naïve CD4+T cells without co-culture served as the negative control (NC). The results indicate that BAP31^−/−^ significantly enhanced the activation markers (CD25, CD69) of naïve CD4^+^T cells (Figure 2A), as well as provide a statistical summary of cell numbers from flow cytometry data (Figure 2B). To further confirm the impact of BAP31-MCKO on CD4^+^T cell activation, peritoneal macrophages (PEMs) from mice were collected, and co-culture experiments were conducted with naïve CD4^+^T cells under the same conditions as BMDMs. The FACS analysis results demonstrated that CD4^+^T cells, when subjected to co-culture experiments with BAP31^−/−^/PEMs, exhibited elevated levels of the T-cell activation markers CD25 and CD69 (Appendix A). Since we observed BAP31-MCKO facilitate CD4^+^T cell activation in vivo, we further validated the function of BAP31 utilizing in vitro cell lines (RAW264.7 cells co-cultured with EL4). BAP31 knockout stable transfected RAW264.7 cells were obtained using a lentivirus system with a fluorescent label, and monoclonal cell lines were established through antibiotic screening (Appendix A). The knockout efficiency of BAP31 was confirmed at both the protein and mRNA levels (Appendix A). We co-cultured RAW264.7 cells and EL4 cells for 72 h to detect the effect of BAP31 knockout of RAW264.7 on the activation of EL4 cells. The findings of this study indicate that the absence of BAP31 in RAW264.7 cells resulted in a significant increase in the expression of CD25 and CD69 molecules in EL4 cells at the protein level (Figure 2C), as well as a statistically significant change in cell numbers (Figure 2D). Additionally, the mRNA levels of T cell activation markers were found to be consistent with the observed protein expression levels, as confirmed by quantitative real-time PCR analysis (Figure 2E).

### 2.3. Macrophage BAP31 Knockout Influences CD4^+^T Cell Differentiation

Naïve CD4^+^T cells can differentiate into different subsets and perform different functions. The differentiation and development of CD4^+^T cells in the spleen and thymus of mice with a macrophage-specific knockout of BAP31 were observed. Our observations revealed that the absence of BAP31 in mouse splenocytes and thymocytes had varying effects on the differentiation of CD4^+^T cells, leading to a reduction in Th1 (IFN-γ) and Th2 (IL-4) populations, as well as an increase in the Th17 (IL-17A) population (Figure 3A). Furthermore, we performed statistical analysis to summarize the cell numbers obtained from flow cytometry (Figure 3B). The existing co-culture system in vitro can achieve the maturation and differentiation of CD4^+^T cell subsets [27,28]. The co-cultures were conducted following the aforementioned protocol for CD4^+^T cell proliferation. Additionally, we utilized flow cytometry to assess the differentiation of CD4^+^T cells. Notably, the absence of BAP31 had a significant impact on the differentiation of naïve CD4^+^T cells, resulting in a decrease in the expression of Th1 (IFN-γ) and Th2 (IL-4), while Th17 (IL-17A) expression was increased (Figure 3C–E). Furthermore, we performed statistical analysis to summarize the cell numbers obtained from the flow cytometry data (Figure 3F). To investigate this further, we co-cultured RAW264.7 cells (BAP31) and BAP31 knockout RAW264.7 cells (sh-BAP31) with EL4 cells for a duration of 72 h under various conditions, including control, LPS, and IL-4 stimulation. EL4 cells were collected in order to investigate their differentiation. The findings indicated a decrease in the performance of Th1 (IFN-γ) and Th2 (IL-4) in sh-BAP31, while Th17 (IL-17A) levels increased at the RNA levels (Appendix A).

### 2.4. Deficiency of Macrophage BAP31 Influences Antigen Presentation

In the process of T cell interaction with macrophages, the recognition of the macro-phage surface antigen peptide–MHC complex by T cells facilitates the transmission of antigen information, leading to the initiation of an immune response [7]. In order to evaluate the expression of antigen-presenting molecules (CD86, MHC-II, CD80), flow cytometry was employed to analyze mouse bone marrow-derived macrophages from different groups (control, LPS, and IL-4). The results indicated that the levels of macrophage antigen-presenting molecules (CD86, MHC-II, CD80) were higher in the BAP31^−/−^ group compared to the BAP31^+/+^ group, with a notable increase in the MHC-II molecule (Figure 4A). Flow cytometric analysis demonstrated a statistically significant elevation in the mean fluorescence intensity in the BAP31^−/−^ group compared to the BAP31^+/+^ group. (Figure 4B). Similarly, the results obtained from RAW264.7 cells demonstrated that the expression of macrophage antigen-presenting molecules (MHC-II molecules) was affected by BAP31: macrophage knockdown of BAP31 resulted in an upregulation of MHC-II expression at the molecular level (Figure 4C). Flow cytometric analysis revealed an increase in the mean fluorescence intensity in the RAW264.7 cells following the knockdown of BAP31, in comparison to the RAW264.7 cells (Figure 4D). This suggests that the MHC-II molecule of macrophages may play a role in the impact of BAP31 on the differentiation of CD4^+^T cells.

### 2.5. BAP31-MCKO Influences T Cell Differentiation by Upregulation of MHC-II Molecule

We speculate that the MHC-II molecules in macrophages are involved in the effect of BAP31 on CD4^+^T cell differentiation. To investigate this further, we introduced a monoclonal antibody targeting MHC-II into the aforementioned co-culture system to inhibit its influence, and subsequently assessed the differentiation of naïve CD4^+^T cell subsets. The results of this study indicate that the use of blocking antibodies to inhibit MHC-II has a notable effect on the differentiation of CD4^+^T cells. CD4^+^T cells (Th1, Th2 and Th17) decreased both in percentage and number after MHC-II antibody blockade (Figure 5A–C). Furthermore, we performed statistical analysis to summarize the cell numbers obtained from flow cytometry (Figure 5D). Specifically, the blocking of MHC-II resulted in the elimination of the decrease in differentiation of Th1 (IFN-γ) and Th2 (IL-4), as well as the increase in differentiation of Th17 (IL-17A). These findings support our hypothesis that BAP31 plays a crucial role in regulating the expression of MHC-II molecules in macrophages, thereby affecting the activation of CD4^+^T cells and influencing their differentiation.

## 3. Discussion

Macrophages, as important antigen-presenting cells, are capable of processing and presenting antigens to T cells, thereby participating in the immune response. The interaction between macrophages and T cells builds a bridge between innate and adaptive immune responses [29]. Previous investigations conducted by our research team have demonstrated the involvement of BAP31 in the regulation of macrophages and T cells [23,30]. In the current study, we utilized mice with a macrophage-specific knockout of BAP31 to investigate the impact of BAP31 on the activation of CD4^+^T cells by macrophages.

The thymus and spleen, which are primary lymphoid organs, play a critical role in the development and maturation of T cells [31,32]. Changes in the spleen and thymus in-dices in mice reflect their immune capacity [33]. We studied the size of the spleen and thymus in BAP31-MCKO mice and found that all BAP31^−/−^ mice had larger spleens and thymuses than their BAP31^+/+^ littermates. The thymus and spleen are important immune organs in body. The thymus is an important central immune organ composed of the cortex and medulla. The spleen represents an important secondary lymphoid organ harboring many immune cells [31,32]. We examined the number of main immune cells in the thymus and spleen of mice using flow cytometry. We observed that BAP31-MCKO had no discernible impact on the development of CD8 T cells, B cells, and dendritic cells, but did affect the frequency of CD4 T cells. Upon TCR activation triggered by antigen-presenting cells, naive CD4^+^T cells differentiate into distinct Th lineages in the context of combinations of transcription factors and cytokines [34]. We evaluated the expression of molecules related to CD4^+^T cell activation and differentiation in the spleen and thymus under physiological conditions. Each Th lineage can produce a cytokine with the potential to play a positive feedback role in promoting differentiation: IFN-γ for Th1, IL-4 for Th2 and IL-17 for Th17 [12,35]. The observed disparities in mRNA levels of genes in the spleen and thymus of BAP31^−/−^ mice suggested that macrophages play a role in the activation and differentiation of CD4^+^ T cell lineages. To investigate the impact of BAP31-deficient macrophages on T cells in vitro, co-culture systems involving BMDMs/CD4^+^T cells and RAW264.7 cells/EL4 cells were established. By employing a combination of staining techniques to assess the expression of CD25 and CD69 in mouse CD4^+^ T cells, we examined the activation of these cells under various stimulus conditions. The current study demonstrated that deficiency of BAP31 in BMDMs or RAW264.7 cells leads to an upregulation of CD25 and CD69 expression in CD4^+^T cells and EL4 cells. CD69 is an early T-cell activation marker, as its presence on T cells is followed by CD25 and at later stages of activation by MHC class II, HLA-DR [4].CD25 expression was rather weak compared to CD69 expression in our assay. These findings suggest that BAP31 exerts modulatory effects in the activation of CD4^+^T cells induced by macrophages, specifically in regulating the early stage of T cell activation.

Naïve CD4^+^ T cells are able to differentiate into various subtypes and migrate to different compartments within the lymphoid or peripheral regions [34]. The Th1 program is initiated by IFN-γ, which activates the lineage-specific transcription factor T-bet [36,37]. GATA-3 is a transcription factor necessary for the differentiation of the Th2 lineage of effector CD4^+^T cells [38]. RORγt, as a specific transcription factor of Th17, promotes the CD4^+^ T cell lineage differentiation after positive selection [39]. T cell differentiation is initiated by signals from the T cell antigen receptor, costimulatory molecules and cytokine receptors [40]. Macrophages can enhance antigen presentation ability by increasing MHC-II and costimulatory molecules CD80 and CD86 labeling on the cell surface [41]. LPS/IL-4 stimulation influenced antigen presentation ability compared with the unstimulated macrophages. The antigen presentation ability of macrophages (including MHC-II and costimulatory molecules CD80 and CD86) is involved in CD4+T cell activation. Despite the functional differences between M0, M1, and M2 macrophage subsets, it is generally assumed that the expression and function of broadly defined housekeeping genes, which regulate fundamental cellular processes, are consistent or similar across all of these macrophage subtypes [42]. In the present study, we successfully demonstrated that the absence of BAP31 in BMDMs or RAW264.7 cells hinders the development of Th1 and Th2 cells, while promoting the differentiation of Th17 cells. Our findings indicate that BAP31 plays a modulatory role in the differentiation of CD4^+^T cells induced by macrophages. The diverse subsets of CD4^+^T cells, including Th1, Th2 and Th17 cells, are crucial in providing protection against a wide range of pathogens [43]. Specifically, Th1 cells are primarily responsible for defending the host against intracellular pathogens such as viruses, protozoa, and bacteria, Th2 cells play a critical role in mediating immune responses against extracellular parasites, and Th17 cells are essential for orchestrating immune responses to extracellular bacteria and fungi [44]. Different Th lineages rely on distinct signaling pathways for their development. More studies are needed to clarify the mechanisms of BAP31 modulation of CD4^+^T cell differentiation in macrophages.

TCR binding of the MHC class II peptide complex, along with the interaction of appropriate costimulatory molecules, activates naïve T cells, which then differentiate into different Th subtypes, depending on the cytokine environment [45,46]. Major histocompatibility complex (MHC) class II molecules are ligands for CD4^+^T cells and are critical for initiating the adaptive immune response [47,48]. CD80 and CD86 represent a dominant costimulatory pathway that plays a critical role in CD4^+^T cell activation and proliferation [49]. Drawing upon previous research on the functional and trafficking aspects of BAP31, in conjunction with the antigen presentation properties of macrophages, it is imperative to further explore the expression of antigen-presenting molecules (MHC-II, CD80, CD86) in macrophages. The findings of our study indicate that the absence of BAP31 in BMDMs or RAW264.7 cells leads to an upregulation of antigen-presenting molecules, particularly MHC-II. This observation was further validated through a co-culture experiment, where the use of MHC-II neutralizing antibody successfully inhibited MHC-II activity in the culture. These results provide initial evidence suggesting that BAP31 plays a role in the activation and differentiation of CD4^+^T cells by influencing the antigen presentation function of macrophages.

The findings of our study are summarized in a schematic diagram (Figure 6). BAP31 is implicated in the activation and differentiation of CD4^+^T cells through its impact on the MHC-II of macrophages, thereby facilitating the interaction between macrophages and CD4^+^T cells. Further investigations are warranted to elucidate the precise molecular mechanism underlying the relationship between BAP31 expression in macrophages and the differentiation of CD4^+^T cells, as well as the influence of BAP31 modulation on the differentiation of the different T helper subsets. 

## 4. Materials and Methods

### 4.1. Mouse Models

BAP31-MCKO mice were generated by crossing BAP31^flox/flox^ mice with Lyz2-Cre mice on a C57BL/6 background. Experiments were conducted when the mice were 8–12 weeks old. The mice were bred and maintained under specific pathogen-free conditions at Northeastern University, following the guidelines set forth by the institution. The mice were housed at a temperature of 23 °C with a 12 h light/12 h dark cycle. All experiments were performed in accordance with a protocol approved by the Institutional Animal Care and Use Committee.

### 4.2. Bone Marrow-Derived Macrophages

After euthanasia, the mice were subjected to alcohol immersion disinfection for a du-ration of 3 min. Following this, the femurs and tibias were aseptically separated. Subsequently, bone marrow cells were obtained by flushing 1640 medium through the bone cavities of the femurs and tibias using a syringe. The collected bone marrow cells were then treated with a red cell lysis buffer to remove red blood cells. Bone marrow-derived macrophages (BMDMs) were obtained from these bone marrow cells and cultured in 1640 medium (Gibco, New York, NY, USA) supplemented with 10% fetal bovine serum, streptomycin (100 U/mL), penicillin (100 U/mL) and 30% L929-conditioned medium for a period of 7 days at 37 °C under 5% CO_2_ atmosphere.

### 4.3. Cell Culture

The macrophage cell line (RAW264.7) and a CD4^+^T cell line (EL4) were maintained in our laboratory. RAW264.7 was cultured in DMEM medium (Gibco), and EL4 was cultured in RPMI 1640 medium (Gibco) supplemented with fetal bovine serum, 100 U/mL penicillin, and 0.1 mg/mL streptomycin. Cells were routinely cultured in a humidified incubator at 37 °C under 5% CO_2_ atmosphere.

### 4.4. Knockout BAP31 Stable Transfected RAW264.7 Cells

Lentivirus viruses were generated through the transfection of 5 µg of plasmid (2 µg PL/shRNA/GFP-mouse-BAP31, 1.5 µg PMD2G, and 1.5 µg PSPAX2) and 10µL lipo6000 Transfection Reagent (Beyotime Biotechnology, Shanghai, China) into the 293 T cell line. After 48 h of transfection, the viral supernatant was collected, filtered using a 0.45 μm filter (Millipore, Burlington, MA, USA), and subsequently used for transduction of RAW264.7 cell lines. To obtain stably transfected cells, cells were selected using 10 µg/mL Blasticidin (BSD). Fluorescence microscopy (Leica, Wetzlar, Germany) and Western blotting were performed to determine the knockdown efficiency of RAW264.7 cells.

### 4.5. Co-Culture of Macrophages with Naïve CD4^+^T Cells

Spleens were isolated from C57BL mice, and a single cell suspension was prepared. Naïve CD4^+^T cells were obtained from spleen cells through magnetic bead sorting (Miltenyi Biotec, Bergisch Gladbach, Germany). The purity of the sorted cells was assessed immediately using flow cytometry (BD LSRFortessa). Bone marrow-derived macrophages were stimulated with LPS (100 ng/mL) or IL-4 (20 ng/mL) for 24 h. The macrophages were then divided into three groups: control, LPS, and IL-4. Naïve CD4^+^T cells obtained through magnetic sorting at a ratio of 10:1 were added to the macrophages (containing CD3 at a concentration of 3 µg/mL) for co-culturing in a contact system in flat-bottomed 6-well plates for a duration of 72 h.

### 4.6. Western Blotting

Cells were collected and washed twice with PBS. Cell proteins were obtained by RIPA lysis buffer (8 M urea, 2 M thiourea, 3% SDS, 75 mM DTT, 0.05 M Tris-HCl [pH 6.8], and 0.03% bromophenol blue) for 30 min on ice. The resulting mixture was centrifuged at 12,000× *g* for 15 min, and the proteins were subsequently separated by electrophoresis using a 12% SDS-PAGE gel. Protein bands were transferred to PVDF membranes (Bio-Rad, Hercules, CA, USA), and then membranes were blocked in 5% BSA/TBST for 1 h at room temperature. Membranes were incubated with primary antibody (1:1000 dilution) with rotation at 4 ℃ overnight and horseradish peroxidase conjugated secondary antibody at room temperature for 1 h. Protein detection was performed with ECL reagent (Thermo, Waltham, MA, USA).

### 4.7. Flow Cytometry Analysis

Cells were washed twice with cold PBS and resuspended for antibody staining at 1×10 ^6^ cells/100 uL in 1% BSA/PBS. Following blocking of Fc receptors with 1 ug/mL of Fc blocker (CD16/32, BioLegend, San Diego, CA, USA) for 10 min at room temperature (RT), single-cell suspensions were stained on ice for 30 min with cell type-specific fluorophore-conjugated antibodies. Cells were subsequently washed twice in PBS, fixed with 2% paraformaldehyde, permeabilized with 0.1% saponin and stained with intracellular anti-cytokine antibody for 45 min on ice. After the cells had been washed twice, the treated cells were detected using a flow cytometer (BD C6). Antibodies from eBioscience—CD3 (145-2C11), CD4 (GK1.5), B220 (RA3-6B2), CD8 (53-6.7), CD69 (H1.2F3), IFN-γ (XMG1.2), IL-4 (11B11), IL-17A (eBio17B7), F4/80 (BM8); from BioLegend—CD25 (PC61); from BD—CD11b (M1/70), MHC-II (2G9), CD80 (16-10A1), CD86 (GL1) CD11c (HL3). Data analysis performed using FlowJo software (Version 10).

### 4.8. Quantitative Reverse Transcription PCR

The total RNA was extracted using RNA extraction kit (Takara, Kusatsu, Japan) according to the manufacturer’s instructions. The cDNA was prepared using One-Step RT-PCR System (Invitrogen, Waltham, MA, USA) and then used for real-time PCR. A quantitative real-time PCR was carried out using SYBR Green PCR Master Mix (Promega, Madison, WI, USA) on a BioRad CFX384 real-time PCR machine. The primer sequences for BAP31, CD25, CD69, T-bet, IFN-γ, GATA3, IL-4, RORγt and IL-17A are presented as follows: BAP31 (forward: 5′-ATG AGT TTG CAG TGG ACT ACA GTT G-3′; reverse: 5′-CTC CTC CTT CTT AGC TGA GGG AC-3′), CD25 (forward: 5′-GCT TGC TGA TGT TGG GGT TT-3′; reverse: 5′-ATG GTG CCG TTC TTG TAG GA-3′), CD69 (forward: 5′-GTG GTC CTC ATC ACG TCC TT-3′; reverse: 5′-ACT TCT CGT ACA AGC CTG GG-3′), T-bet (forward: 5′-CCC ATC CCT TCC CTG TAT-3′; reverse: 5′-GTC CAT TCT CCG TTC TCC A-3′), IFN-γ (forward: 5′-ATG AAC GCT ACA CAC TGC ATC-3′; reverse: 5′-CCA TCC TTT TGC CAG TTC CTC-3′), GATA3 (forward: 5′-TCC AGT CCT CAT CTC TTC AC-3′; reverse: 5′-GTC TCC AGC TTC ATG CTATC-3′), IL-4 (forward: 5′-GGT CTC AAC CCC CAG CTA GT-3′; reverse: 5′-GCC GAT GAT CTC TCT CAA GTG-3′), RORγt (forward: 5′-CCG CTG AGA GGG CTT CAC-3′; reverse: 5′-TGC AGG AGT AGG CCA CAT TAC A-3′), IL-17A (forward: 5′-GCC CTC AGA CTA CCT CAA CCG-3′; reverse: 5′-GTC CAG CTT TCC CTC CGC ATT-3′).

### 4.9. Statistical Analyses

All statistical analyses were performed using GraphPad Prism version 7.0 for Windows (GraphPad Software, La Jolla, CA, USA). Statistical comparison was performed using two-tailed Student’s *t* test. Symbols indicate statistical significance as follows: * *p* < 0.05, ***p* < 0.01, and *** *p* < 0.001.

## 5. Conclusions

In conclusion, our findings demonstrated that downregulation of BAP31 in macro-phages regulated the activation and differentiation of CD4^+^T cells through the MHC class II pathway. Taken together, our findings provide evidence that the downregulation of BAP31 in macrophages has significant implications for the immune response.

## Figures and Tables

**Figure 1 ijms-24-13476-f001:**
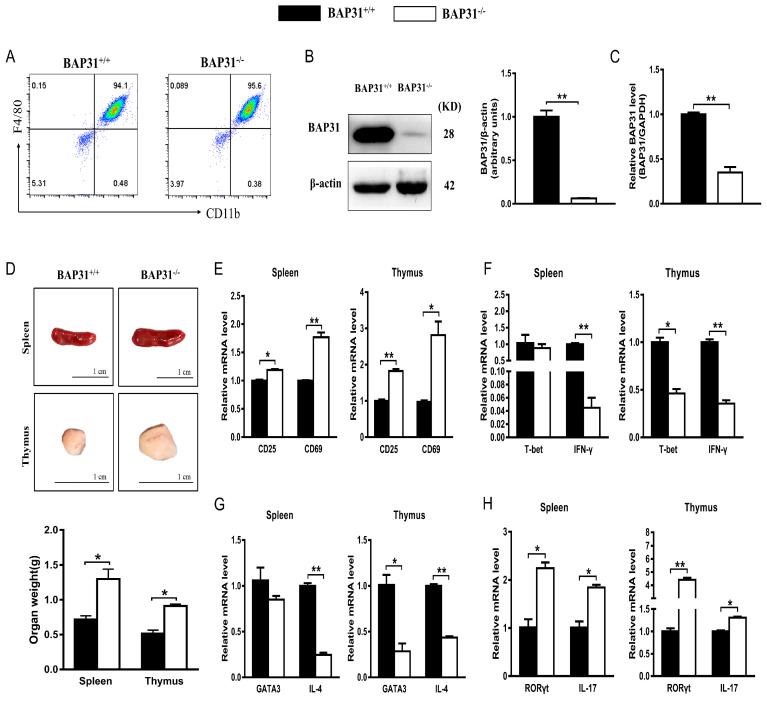
BAP31 is involved in the development of spleen and thymus in BAP31-MCKO mice. (**A**) Flow cytometry-detected BMDM (CD11b^+^F4/80^+^) purity of BAP31^flox/flox^ (BAP31^+/+^) and BAP31^flox/flox^Lyz2-cre (BAP31^−/−^) mice. (**B**) Western blotting analysis of knockout efficiency of BAP31 from BMDMs (*n* = 3). Relative protein expression is expressed as the ratio of BAP31 to β-actin. (**C**) Real-time PCR analysis of knockout efficiency of BAP31 from BMDMs (*n* = 3). Relative BAP31 expression was normalized by GAPDH expression. (**D**) Statistics of the weight of spleen and thymus from mice (*n* = 5). (**E**) RT-qPCR analysis of genomic RNA of T cell activation molecules (CD25, CD69) from spleen and thymus. Total splenocyte and thymocyte RNA from BAP31^flox/flox^ mice (BAP31^+/+^) and BAP31^flox/flox^Lyz2-cre mice (BAP31^−/−^) (*n* = 3). (**F**) RT-qPCR analysis of Th1 cell transcription factor (T-bet) and cytokine (IFN-γ) from spleen and thymus (*n* = 3). (**G**) RT-qPCR analysis of Th2 cell transcription factor (GATA3) and cytokine (IL-4) from spleen and thymus (*n* = 3). (**H**) RT-qPCR analysis of Th17 cell transcription factor (RORγt) and cytokine (IL-17) from spleen and thymus (*n* = 3). Relative gene expression was measured by qRT-PCR and normalized by GAPDH. * *p* < 0.05. ** *p* < 0.01.

**Figure 2 ijms-24-13476-f002:**
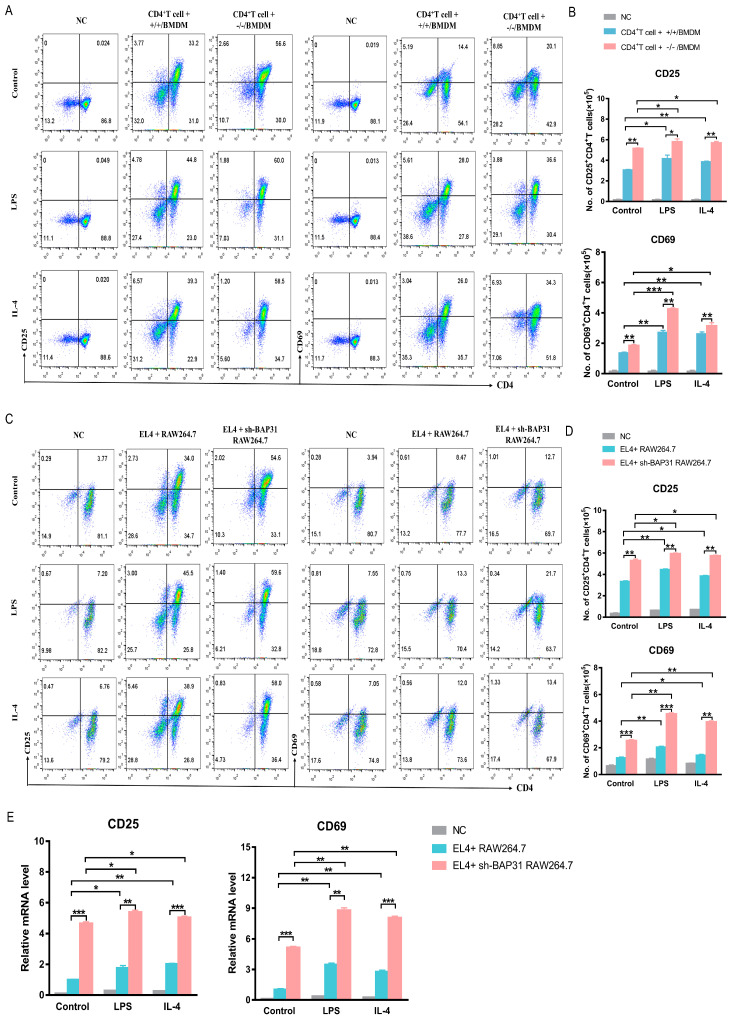
BAP31-MCKO facilitates CD4^+^T cell activation. (**A**) Flow cytometry detected T cell activation markers (CD25, CD69) of naïve CD4^+^T cells co-cultured for 72 h with bone marrow-derived macrophages divided into control, LPS and IL-4 groups. Naïve CD4+T cells without co-culture served as the negative control (NC). (**B**) Statistical bar charts showing the cell numbers of CD4^+^T cell + +/+/BMDM and CD4^+^T cell + −/−/BMDM group (*n* = 3). (**C**) Flow cytometry detected T cell activation markers (CD25, CD69) of EL4 cells co-cultured for 72 h with RAW264.7 cells divided into control, LPS and IL-4 groups. EL4 cells without co-culture served as the negative control (NC). (**D**) Statistical bar charts showing the cell numbers of EL4 + RAW264.7 and EL4 + sh-BAP31 RAW264.7 group (*n* = 3). (**E**) RT-qPCR analysis of T cell activation markers (CD25, CD69) of EL4 cells co-cultured for 72 h with RAW264.7 cells divided into control, LPS and IL-4 groups (*n* = 3). Relative gene expression was measured by qRT-PCR and normalized by GAPDH. * *p* < 0.05. ** *p* < 0.01. *** *p* < 0.001.

**Figure 3 ijms-24-13476-f003:**
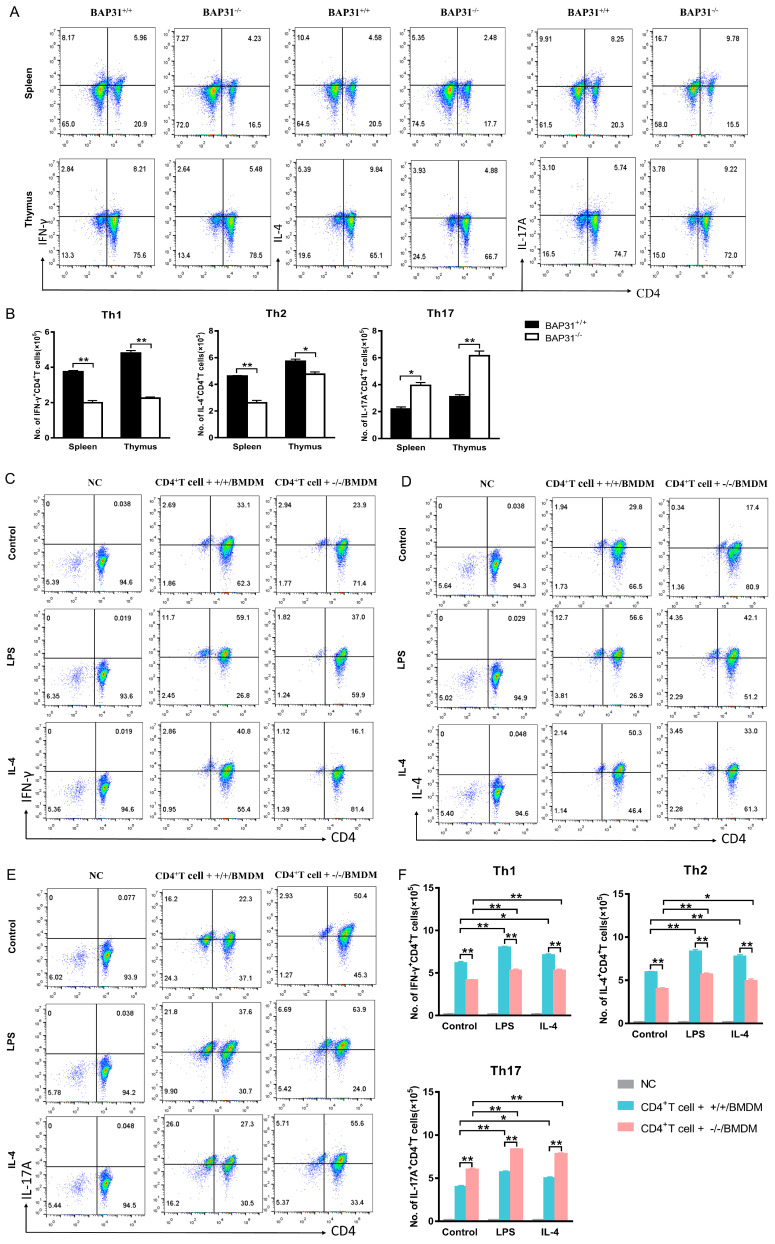
Macrophage BAP31 knockout influences CD4^+^T cell differentiation. (**A**) Flow cytometry detected differentiation of CD4^+^T cell subsets Th1 (IFN-γ), Th2 (IL-4), Th17 (IL-17A) in splenocytes and thymocytes from BAP31^flox/flox^ mice (BAP31^+/+^) and BAP31^flox/flox^fLyz2-cre mice (BAP31^−/−^). (**B**) Statistical bar charts showing the cell numbers of BAP31^+/+^ and BAP31^−/−^groups (*n* = 3). (**C**–**E**) Flow cytometry detected differentiation of CD4^+^T cell subsets Th1 (IFN-γ), Th2 (IL-4), and Th17 (IL-17A). Naïve CD4^+^T cell co-cultured with bone marrow-derived macrophages for 72 h compared with control, LPS and IL-4 groups. Naïve CD4^+^T cells without co-culture served as the negative control (NC) (**F**) Statistical bar charts showing the cell numbers of CD4^+^T cell + +/+/BMDM and CD4^+^T cell + −/−/BMDM group (*n* = 3). * *p* < 0.05. ** *p* < 0.01.

**Figure 4 ijms-24-13476-f004:**
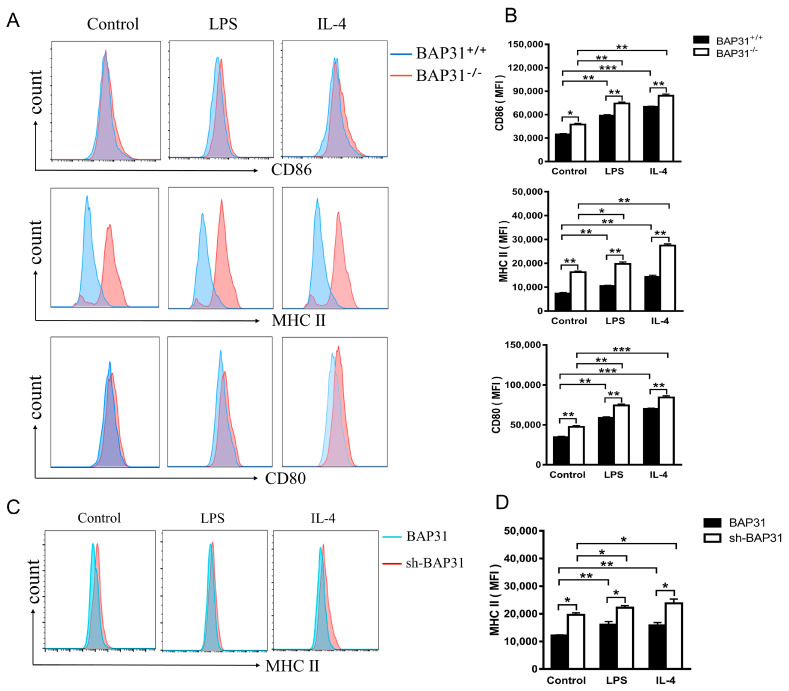
BAP31 influences macrophage antigen. (**A**) Flow cytometry detected antigen-presenting molecules (CD86, MHC-II, CD80) of bone marrow-derived macrophages divided into control, LPS and IL-4 groups. (**B**) Quantification of the mean fluorescence intensity (MFI) of the antigen-presenting molecules (CD86, MHC-II, CD80) by flow cytometric analysis (*n* = 3). (**C**) Flow cytometry detected antigen-presenting molecules (MHC-II) of RAW264.7 cells divided into control, LPS and IL-4 groups. (**D**) Quantification of the mean fluorescence intensity (MFI) of the antigen-presenting molecule MHC-II by flow cytometric analysis (*n* = 3). * *p* < 0.05. ** *p* < 0.01. *** *p* < 0.001.

**Figure 5 ijms-24-13476-f005:**
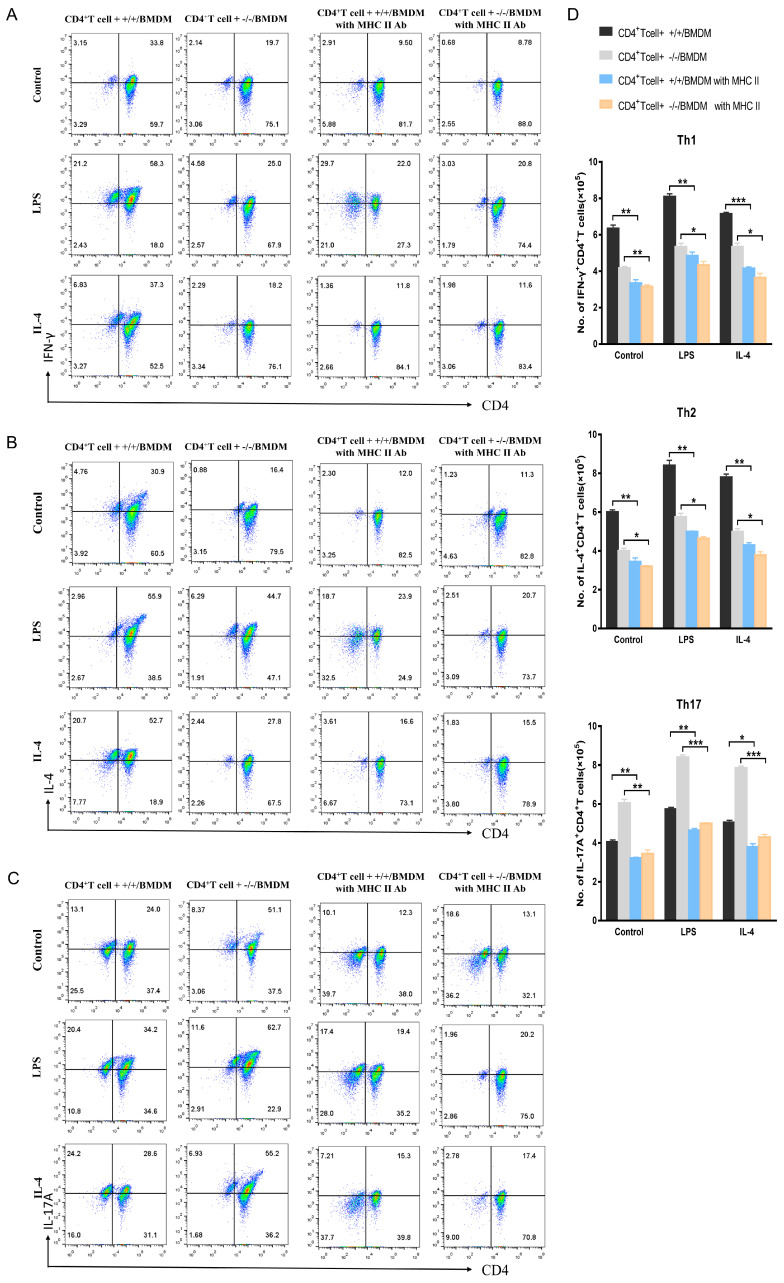
BAP31-MCKO influences T cell differentiation by regulating MHC-II expression levels. (**A**–**C**) Flow cytometry detected CD4^+^T cell differentiation of Th1 (IFN-γ), Th2 (IL-4) and Th17 (IL-17A) under four different conditions: CD4^+^T cells + +/+/BMDMs, CD4^+^T cells + −/−/BMDMs, CD4^+^T cells + +/+/BMDMs with MHC-II Abs, CD4^+^T cells + −/−/BMDMs with MHC-II Abs. (**D**) Statistical bar charts showing the cell numbers of CD4^+^T cells + +/+/BMDMs, CD4^+^T cells + −/−/BMDMs, CD4^+^T cells + +/+/BMDMs with MHC-II Abs, CD4^+^T cells + −/−/BMDMs with MHC-II Abs. * *p* < 0.05. ** *p* < 0.01. *** *p* < 0.001. *n* = 3 for each group.

**Figure 6 ijms-24-13476-f006:**
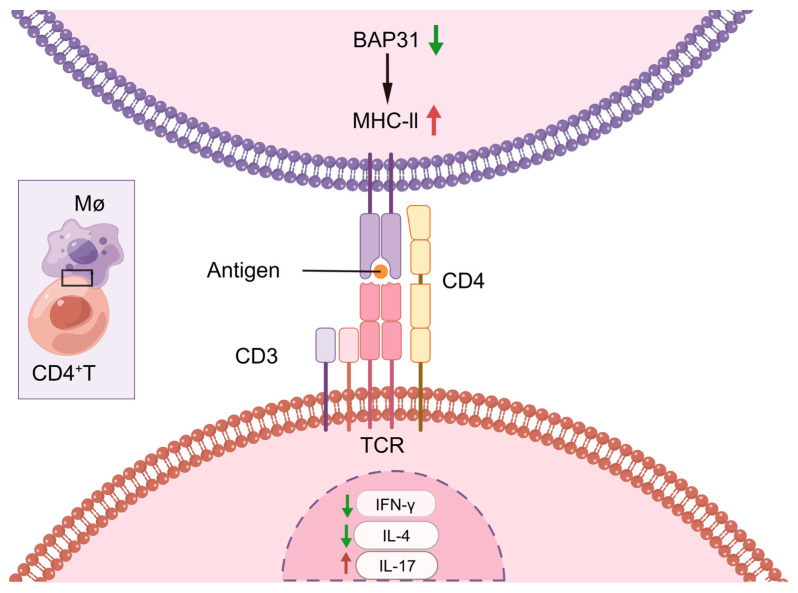
Schematic diagram. BAP31 knockdown led to an increase in MHC-II expression in macrophages, which promoted an increase in CD4^+^T cell activation levels. Ultimately, this series of events impacted the differentiation of CD4^+^T cells, resulting in a decrease in Th1 and Th2 cells and an increase in Th17 cells.

## Data Availability

The data that support the findings of this study are available from the corresponding author upon reasonable request.

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
