# Peer review of "BAP31 Knockout in Macrophages Affects CD4+T Cell Activation through Upregulation of MHC Class II Molecule"

_ijms, 2023, doi:10.3390/ijms241713476_

Round 1
Reviewer 1 Report
This study is interesting but need a lot of improvements. Please see the comments bellow
Major Comments
1. Line 76 to 80 is not clear what authors want to say in this sentence. Cd4 t and macrophages are entirely two different type of cells then how they study CD4T cells in macrophages I do not understand.
2. How BAP31 is involved in the development of spleen and thymus in BAP31 KO?
3. In bot Fig1 B and C Authors should mentioned the macrophages source. Relative expression over, what? For both protein and mRNA.
4. Figure 1E -H data comes from spleen and thymus, which contains other cells too. Then how authors make the conclusion that BAP31 knockout mice were found to induce the activation of T cells in the spleen and thymus. Until and unless these were performed in isolated cells.
5. Is the deletion of BAP31 in macrophages affects developmental of other cells or only activation?
6. What is source of CD4+ T Cells? Is isolated from wild type or knockout mice spleen. Is any difference between the total no. of spleen and Cd4 t cells among the groups?. What is the ratio of T cells and macrophages during co-culture?
7. Why control have activation?
8. Why in control co-culture have activation explain it. Show the lentivirus transfection data either by microscopy or flow. Why authors doing the cell line experiment if they have primary cells? Shows the co-culture data from primary with cell lines for both macrophage and T cells.
9. Why activation markers are elevated in control group.
10. For better underestimating, provide the total no of cells, respective subtype of t cells and relative cytokines from both spleen and thymus.
11. In co-culture of Figure 3 authors did not mentioned these cells comes from same source or different, without knowing that it is hard to say that BAP31 -/- macrophages have role in CD4 cells activation. Provides the cells no. instead of frequency. Did authors checked CD25 and CD69 in Th1, Th2 and Th17 cells.
12. To calm the BAP31 deficient macrophages involved in T cells development differentiation and activation, authors would performed experiment with LPs and IL4 stimulation in CD4 and compared with co-culture conditions from both knockout and wild time mice.
13. Fig4 A correct the labeling which one is CD80/CD86. For MHCII, CD80 and CD86 expression if control group have already expressing these then what is the purpose of using LPS and IL4 stimulation.
14. Are T cells stimulated with BAP3 KO or WT supernatant?
15. Blacking antibody experiments seem not working. Provide the quantification data with better images.
Minor Comments
1. CD4+ T cells in presence of the macrophages
2. Include the sample size in figure legend
3. Supplementary figure 1A, 2 and 3 are missing.
Minor editing needed.
Reviewer 2 Report
This work about the action of BAP31 macrophages to increase the expression of CD4+ T cells, the role that may be crucial of BAP-31 macrophages and deepen the mechanisms of action of macrophages. Its impact on immune processes. Tables and Figures in this work are well developed and easy to understand.
The work is good and detailed,may be of great use, although I think you should deepen in the next research on BAP31.
Round 2
Reviewer 1 Report
Authors have answered my some criticism but not all. Although I asked valuable question which were important for the study and reader as well. Please see the comment below.
In Point 3: I asked relative expression of what? but seems authors did not understand, my point was to know that it was relative expression of b-actin or some other house keeping gene (in case of WB protein and m-RNA)? if yes please mentioned it in the Y-axis as well as figure legend.
In Point4: Without study other cells by flow and making a conclusion that CD4+T cells are activated by BAP31 Knockout macrophages, only because of authors study T-cells activation and not want to study other equivalent cells, doses not make sense. Authors should provide at least some cells which does not affected by this knockout through flow.
In Point5: Authors making a lame excuse to know the influence of other cells in absence of BAP31. They should check in this manuscript.
In Point7, 8 and 9 : Authors saying that T cell activation requires a costimulatory signal provided by T cell CD28 association with CD80/86 on the APC[5]. Every immunologist know this. My point is if control have activation then what is the purpose of using LPS and IL4 for activation in this study so better provide the negative control which showed no activation. In these section authors mentioned co-culture condition and showed stimulated with LPS/IL4 have the same level of activation as control. Which do not understand so please explain it.
In Point 12: Just saying this "In the absence of macrophages, the activation of CD4+T cells solely through LPS or IL-4 stimulation becomes challenging due to the absence of co-stimulatory molecules and cytokines provided by macrophages". It is not convincing showed the CD4 stimulatory data with LPS and IL4 then make this conclusion otherwise this is the only speculated outcome.
Not needed
Round 3
Reviewer 1 Report
I recommend it for publication.